# Preparation and Evaluation of Topically Applied Azithromycin Based on Sodium Hyaluronate in Treatment of Conjunctivitis

**DOI:** 10.3390/pharmaceutics11040183

**Published:** 2019-04-15

**Authors:** Qian Chen, Chun Yin, Jiang Ma, Jiasheng Tu, Yan Shen

**Affiliations:** 1Department of Pharmaceutics, China Pharmaceutical University, 24 Tong Jia Xiang, Nanjing 210009, China; 1621010133@stu.cpu.edu.cn (Q.C.); yinchun8124@gmail.com (C.Y.); 2Center for Research Development and Evaluation of Pharmaceutical Excipients and Generic Drugs, China Pharmaceutical University, 24 Tong Jia Xiang, Nanjing 210009, China; 3School of Biomedical Sciences, Faculty of Medicine, the Chinese University of Hong Kong, Hong Kong 999077, China; majoriea@163.com

**Keywords:** azithromycin, sodium hyaluronate, bioadhesion, eye drops, ocular bioavailability, safety evaluation

## Abstract

Azithromycin (AZI) eye drops containing sodium hyaluronate (SH) were developed to improve the bioavailability of AZI. Interaction between AZI and SH in the AZI-SH formulation was investigated by differential scanning calorimetry, X-ray diffraction, and ^1^H-nuclear magnetic resonance spectroscopy analyses. Moreover, advantages of using SH as an excipient were investigated by comparing physiological properties and pharmacokinetic behaviors of SH-containing AZI eye drops with that of hydroxypropyl methylcellulose (HPMC)-containing formulation. In addition, safety of the developed AZI-SH eye drops was evaluated by in vitro 3-(4,5-dimethyl-2-Thiazyl)-2, 5-diphenyl-2H-tetrazolium bromide assay (MTT assay) and neutral red uptake assay as well as in vivo eye irritation test and acute toxicity test. The results indicated that AZI formed a complex with SH under a slightly acidic condition. The area under the curve (AUC) of AZI in SH-containing formulation was 1.58-fold higher (*P* < 0.01) than that in HPMC-containing formulation due to the interaction between the amine group of AZI and the carboxyl group of SH, despite of the higher viscosity of HPMC-containing formulation. Safety evaluation showed that AZI-SH eye drops caused no obvious eye irritation and acute toxicity. In conclusion, the developed SH-containing AZI formulation possessing advantages of longer retention time and higher drug availability was a promising drug formulation for topical ocular therapy.

## 1. Introduction

The prevalence of conjunctivitis, especially those caused by Chlamydia trachomatis remains very high. It has been reported by World Health Organization (WHO) that there are approximately 300–500 million people who had suffered from trachoma [1]. Azithromycin (AZI) is a new type of semi-synthetic macrolide antibiotic, which shows antibacterial effects to a variety of common pathogenic bacteria and has been used to treat respiratory infections, skin and soft-tissue infections and genital infections [2]. Previous reports have demonstrated that the oral dosage forms of azithromycin were effective for the treatment of eye infections, such as conjunctivitis and other sensitive pathogens [3,4,5]. However, at least 1.0 g of AZI was required for each oral dose to ensure the sufficient concentration of AZI in aqueous humor, tears and conjunctiva coat to exert its inhibitory effects. As a consequence, the high oral dose of AZI might enhance the risk of side effects due to the high concentration of AZI in other normal tissues [6,7]. Therefore, a topical ophthalmic formulation of AZI with advantages of lower dose frequency, more convenience and better compliance is required, which may also facilitate the reduction of the risk of selecting resistant bacteria [8]. AZI salt form was commonly applied to prepare its aqueous ocular formulation owing to the deficient water solubility [9,10]. Nevertheless, the short retention time of effective drugs on the ocular surface has hampered the use of traditional topical ophthalmic formulations. After instillation, the eye’s drainage system was activated by exogenous drugs, and blinking and lachrymation significantly accelerated the excretion thus lowering the efficacy of the drugs. On the other hand, to maintain the effective drug concentration in the tear film, more frequent instillations were required, which might cause side effects and result in poor patient compliance [11].

In order to enhance the persistence of AZI in eyes, various mucoadhesive materials possessing precorneal retention properties have been used to improve ocular drug bioavailability, such as sodium hyaluronate (SH), hydroxyl methylcellulose, polyvinyl alcohol, hydroxypropyl methylcellulose (HPMC), carboxyl methylcellulose and polycarbophil [12,13,14,15,16]. AzaSite^®^, by taking DuraSite as drug delivery system, was an ophthalmic AZI formulation on the market to treat bacterial conjunctivitis. Crosslinking acrylic acid resin was used in this system to stay in contact with conjunctiva to prolong the retention of AZI [17,18]. However, eye irritation has been frequently reported in about 1–2% of patients who received AzaSite^®^ [18]. In the present study, SH with bioadhesive feature and mutual effect was employed in the ophthalmic AZI formulation for the treatment of bacterial infections.

Hyaluronic acid, a kind of natural linear mucopolysaccharide, consists of β-1,4 glucuronic acid and β-1,3 acetylglucosamine disaccharides repeat units. Hyaluronic acid has been widely used as an excipient in various drug delivery systems due to its biocompatibility, safety and biodegradability [19,20]. It is normally existed in the form of sodium salt, SH, which is a highly hydrophilic and negatively charged polymer with biological adhesion effect [21]. The binding sites of hyaluronate have been identified on the corneal endothelium [22], which contributes to the improved bioavailability of hyaluronate-containing ophthalmic formulation. In addition, SH has also been proposed capable of reducing ocular surface toxicity and alleviating symptoms of ocular surface damage [23,24].

In this paper, biological adhesion effect of SH has been demonstrated by investigating the interaction between AZI and SH. The drug retention effect of SH and HPMC was also compared in terms of viscosity and pharmacokinetic properties of AZI eye drops containing HPMC or SH. In addition, the toxicity of AZI eye drops was evaluated by both in vitro and in vivo approaches.

## 2. Materials and Methods

### 2.1. Materials

AZI was bought from Guoguang Pharmaceutical Co., Ltd. (Hangzhou, China). SH (1500kD, Eye-drop Grade) was bought from Freda Biochem Co., Ltd. (Shandong, China). Acetonitrile (HPLC/SPECTRO) was purchased from Tedia Company Inc. (Fairfield, OH, USA). HPMC K100M was obtained from Colorcon (Shanghai, China) and sodium chloride injection was bought from Shandong Huaru Pharmaceutical Co., Ltd. (Shandong, China). All other reagents used in the present study were of analytical grade.

New Zealand albino rabbits (male and female) weighing 2.0–2.5 kg, free of any signs of ocular disease, were purchased from Nanjing Qinglong Mountain Farm (Nanjing, China). All animal experiments were carried out according to the National Institute of Health Guide for the Care and Use of Laboratory Animals and approved by the Animal Ethics Committee of China Pharmaceutical University (SYXK2017-0019, 20 December 2016 to 19 December 2021).

### 2.2. Methods

#### 2.2.1. Degradation Kinetics of AZI

Detailed experimental procedures of degradation kinetics of AZI are included in the Appendix A.

#### 2.2.2. Preparation of AZI-SH and AZI-HPMC Complex Powders

The optimal formulation contained AZI 10 mg, SH (1500 kD) 6 mg, benzalkonium chloride (BZK) 0.01 mg and ultra-pure water 1 mL. The preparation process was divided into two parts (100 mL eye drops as the sample). For the first part, AZI (1 g) was precisely weighed and dispersed in 40 mL ultra-pure water and phosphoric acid (1 mol/L) was slowly added to adjust pH to 5.5 with continuous stirring until the AZI dissolved thoroughly. Then, BZK (1 mg) was added and sodium chloride (2 mol/L) was used to adjust the osmotic pressure to be equivalent to 1.7% sodium chloride solution with the final volume of 50 mL. For the second part, SH (0.6 g) was dissolved into 50 mL ultra-pure water and mixed with the first part.

The preparation process of AZI-HPMC eye drops was also divided into two parts. For the first part, AZI (1 g) was precisely weighed and dispersed into 40 mL ultra-pure water. The dilute phosphoric acid (1 mol/L) was slowly added with rapid stirring until the pH was 5.5. For the second part, HPMC was dissolved into 50 mL hot water at 80–90 °C, and then the solution was stirred until cooling to the room temperature. These two parts were mixed together to obtain the 1% AZI eye drops and the osmotic pressure was adjusted with sodium chloride solution (2 mol/L) with the final volume of 100 mL.

The prepared homogeneous solution of AZI-SH and AZI-HPMC eye drops was filtered through a 0.22 μm filter. The complexes of AZI-SH and AZI-HPMC were obtained from the lyophilized powders of the prepared eye drops.

#### 2.2.3. Differential Scanning Calorimetry (DSC)

Drug-polymer interaction was investigated by differential scanning calorimetry (DSC), which was carried out on a DSC 250 detector (Pyris Diamond TG/DTA, PerkinElmer, Waltham, MA, USA). The investigated samples (5 mg) were sealed into a platinum crucible. DSC scan was recorded in the range of 30 °C to 350 °C at a heating rate of 10 °C/min under a nitrogen purge, and an empty pan was used as a reference. The analyses were performed for pure AZI, SH, AZI-SH, physical mixture of AZI and SH.

#### 2.2.4. X-Ray Diffraction Analysis

X-ray diffraction (XRD) was measured at D8 Advance X-ray Diffractometer (Bruker-AXS company, Karlsruhe, Germany) using Cu Kα radiation (λ = 1.54 Å) and APEX II CCD detector. Diffraction angle 2-Theta scale was measured in range from 3° to 40° with step size 0.02° at speed 4 °/min. Generator was operated at current 40 mA and voltage 40 kV. XRD analysis was performed using AZI, SH, AZI-SH and physical mixture of AZI and SH.

#### 2.2.5. ^1^H-Nuclear Magnetic Resonance Spectroscopy (^1^H-NMR)

The ^1^H-NMR of AZI, SH, AZI-SH and physical mixture of AZI and SH dissolved in the mixture of acetic acid-d6 and D_2_O (2:100, *v*/*v*) were performed with a Varian 500 MHz NMR spectrometer (Bruker, Faellanden, Switzerland) at 25 °C.

#### 2.2.6. Rheological Study

The viscosity of the prepared AZI-SH and AZI-HPMC formulations was determined on a cone (0.8°) and plate geometry viscometer (Brookfield DV-IIIULTRA, Middleboro, MA, USA) equipped with a spindle SC4-16 at 37 ± 1 °C. The viscosity of each formulation was measured under the varied rotational speed increased from 25 to 220 rpm, maintained at the largest speed for 6 s, and decreased from 220 to 25 rpm. The average record of increase and decrease rotational speed was used to evaluate the rheological behavior of AZI-SH, AZI-HPMC formulations, blank SH and blank HPMC. Experiments were done in triplicate.

#### 2.2.7. Azithromycin Quantification by HPLC

The HPLC analysis of AZI was carried out on a DIONEX Ultimate 3000 HPLC system (ThermoFisher, Waltham, MA, USA). Chromeleon 7 software was used for process monitoring, data acquisition and system control. A reversed phase C18 column (Phenomenex luna C18, 5 μm, 150 mm × 4.6 mm) was used for separation and maintained at 40 °C. The mobile phase, composed of phosphate buffer (0.05 mol/L dipotassium phosphate with pH value of 8.2 adjusted by 20% phosphoric acid solution): acetonitrile (45:55) was used at a flow rate of 1.2 mL/min with an isocratic elution. AZI was detected at 215 nm by UV detector. The calibration curve showed a good linearity (*r*^2^ > 0.99). The limit of quantification detection (LOD) was 0.5 μg/mL.

#### 2.2.8. Precorneal Pharmcokinetic Study

New Zealand albino rabbits were housed under standard environmental conditions (25 °C, RH 50%, and 12 h light/dark cycle) with free access to food and water. 10 rabbits were used to determine precorneal pharmacokinetics of AZI. Each rabbit was instilled with 50 μL 1% AZI-SH eye drops prepared according to Section 2.2.2 on the right eye, and 50 μL 1% AZI-HPMC eye drops prepared according to Section 2.2.2 on the left eye, respectively. At the predetermined time points, tears were collected by Schirmer test [25,26]. Briefly, tear samples of both eyes were collected at 5, 10, 20, 30, 45, 60 min after instillation by Schirmer test strips. The amount of tear collected was equal to the gained weight of each strip after sampling. The eyelids were gently held to close during sampling to prevent the loss of eye drops. Subsequently, the Schirmer strip was dried under N_2_ stream and 200 μL of mobile phase was added. The sample was then vortexed to dissolve AZI and centrifuged under 300 g for 10 min. The content of AZI in the supernatant was quantified by HPLC and the amount of AZI was calculated and presented as mg per g of the tears. Pharmacokinetic parameters of AZI, including C_max_, T_max_ and AUC_0-t_, were calculated based on the obtained pharmacokinetic profiles by WinNonlin software.

#### 2.2.9. Safety Evaluation

##### Cytotoxicity Assay

The cytotoxicity of AZI-SH eye drops in vitro was evaluated via 3-(4,5-dimethyl-2-Thiazyl)-2, 5-diphenyl-2H-tetrazolium bromide assay (MTT assay) and Neutral red uptake assay (NRU assay). 3T3-L1 cells were cultured in Dulbecco’s modified Eagle’s medium (DMEM) which contained 10% (*v*/*v*) fetal bovine serum, 100 IU/mL penicillin G and streptomycin, and 2 mM L-glutamine at 37 °C with 5% CO_2_ and air humidified atmosphere. 3T3-L1 cells were exposed to different concentrations of BZK solution, AZI-SH eye drops and commercial AZI eye drops diluted by medium for 24 h. Each concentration was tested in triplicates. DMSO (0.5%) served as the solvent control. The cell viability was then detected by MTT assay. MTT solution (20 μL, 5 mg/mL) was added into each well and incubated for 4 h. Then, the MTT-containing medium was removed and 200 μL of DMSO was added. The absorbance was measured at 490 nm by microplate reader (MD Spectramac M3, San Jose, CA, USA). The cell inhibition rate was calculated using Equation (1):(1)$$\mathrm{Cell}\;\mathrm{inhibition}\;\mathrm{rate}\;(\%)\;=\;(1-\frac{I_{sample}-I_{blank}}{I_{control}-I_{blank}})\times 100\%,$$
where *I_sample_* and *I_control_* are the mean absorbance values of tested group and control group, respectively. *I_blank_* is the absorbance value of the medium.

As for the NRU assay, 3T3-L1 cells were treated with the same concentrations of BZK solution, AZI-SH eye drops and commercial AZI eye drops. After the treatment, cells were incubated with a 200 μL medium containing neutral red dye (50 μg/mL) for 3 h. Cells were then washed three times with (phosphate buffer saline) PBS and the dye was extracted with 200 μL destaining solution (deionised water, ethanol and glacial acetic acid, 49:50:1 *v*/*v*). The absorbance was measured at 540 nm using a microplate reader. Cell viability expressed as percentage of control and medium effective dose (ED_50_), the concentration of 50% reduction in dye uptake, were calculated by the same method described in the MTT assay.

The cytotoxicity of AZI-SH eye drops in vitro was also evaluated via MTT assay toward human corneal epithelial cells (HCE-2). The detailed experiments were listed in Appendix A.

##### In Vivo Eye Rrritation Assessment in Rabbits

10 New Zealand albino rabbits were used to assess the ocular irritation of AZI-SH eye drops according to the Draize eye test [27]. Rabbits were checked to ensure no ocular defects before experiments. Approximately 50 μL of AZI-SH eye drops (1%) was dropped into the conjunctival sac of the right eye of each animal. The lids were held together after administration to prevent loss. The left eye without any treatment was regarded as control. Rabbits were treated with AZI-SH eye drops once a day for two weeks. After the treatment, rabbits’ eyes were washed with saline to remove the residues of the drug [28]. Irritation was scored based on the criterion of Draize test, and the responses were evaluated and listed in Appendix A, expressed as the maximum average score [20,29].

Upon completion of the eye irritation study, rabbits were sacrificed by air embolism under anesthesia. Eyeballs were removed, fixed in 10% formalin solution and embedded in paraffin for further pathological examination. Histologic examinations of cornea, iris, screla and conjunctiva components were conducted with a light microscopy (T8-100, Nikon, Tokyo, Japan). The following aspects of eyeballs were checked, including variation of the ocular surface epithelial cells, edema in lid tissues, presence of inflammatory cells and any other abnormality [30].

##### Acute Toxicity Test

Eight healthy New Zealand albino rabbits without any eye irritation, corneal or conjunctival defects were used for the acute toxicity study. The rabbits were divided into two groups. One group were treated with 3% AZI-SH eye drops (high concentration) in two eyes every four hours for twice, to ensure the total amount was equivalent to 200 times of normal daily dosage, while the other group were given saline into two eyes as the control. The instillation operation was the same as the processes described in eye irritation test in Section 2.2.8. Changes in ocular tissues in animals were evaluated at predetermined time points after treatment.

#### 2.2.10. Statistical Analysis

All values are represented as mean ± standard deviation (SD). The significant differences were evaluated by SPSS using a standard Student’s *t*-test. Values of *P* < 0.05 were regarded as statistical significant.

## 3. Results

### 3.1. Degradation Kinetics of Azithromycin

It is commonly recognized that solubility and stability properties of active pharmaceutical ingredients (APIs) are the major contributing factors for a formulation. The poor solubility of AZI, however, renders it hard to formulate a liquid dosage form. To solve this problem, acidic solutions were proposed to increase the solubility of AZI. It has been reported that the mass solubility of AZI can increase up to 1000 g/L over the range of pH 1–6 at 25 °C and is 100 g/L at pH 7 [31]. The stability of AZI formulation under different pH values should be monitored to achieve a good balance of solubility and stability. In this study, the degradation kinetics and acid-alkali ionization constant of AZI in different phosphate buffer solutions with different pH values were investigated.

As shown in Figure 1A, lnC_AZI_ at predetermined intervals was plotted against time (*t*). The hydrolysis reaction obeyed the first-order kinetics (Equation (2)) owing to the linearity of these plots. The acid-alkali ionization constants (K) in Table 1 were obtained from the slopes of the lines (Figure 1A) as follows:ln*C_AZI_* = lnC_AZI,0_ − *Kt*(2)
where *C**_AZI_* is the concentration of AZI at the *t*(h) and C_AZI,0_ is the initial concentration of AZI.

lnK values were plotted against pH values (Figure 1B). A turn of this curve was observed over pH values of 4.1−8.0, which indicated that AZI was deprotonated to the free alkali form. The lowest point was located at pH 6.0 where AZI possessed the best stability (pH_m_). The degradation of AZI began to increase when pH below or above 6.0. The hydrolysis rate of AZI also took the same tendency. Moreover, the increase of hydrolysis rate was sharp when the pH decreased below 6.0, while only a slight increase of hydrolysis rate occurred with the pH increasing above 6.0. Since there was an ester bond in the structure of AZI, we speculated that the increase in degradation might be caused by the acidic and alkali dual-catalyzed hydrolysis. Taking the results of Box-behnken Design (data not shown) into the consideration, pH 5.5 was chosen for AZI formulation, which was an acceptable pH for ophthalmic preparation [32].

### 3.2. Differential Scanning Calorimetry (DSC)

DSC was used to investigate the interaction between AZI and SH. AZI, SH, physical mixture of AZI and SH and the complex of AZI-SH were examined in the present study. DSC thermograms were depicted in Figure 2A. AZI has a broad melting peak on DSC thermogram, which has been ascribed to the departure of crystalline water simultaneously when melting. AZI from different manufactures were found to exhibit variable thermal, most of them exhibited two typical endotherms [33]. In the DSC thermogram of pure AZI, sharp melting endotherms appeared at 140 °C and 226 °C, which pointed out its melting points and suggested the crystal structure of AZI. In DSC thermogram of pure SH, an endothermic peak at 196 °C and an exothermic peak at 226 °C were observed, which were consistent with previous reports [34]. The thermogram observed from physical mixture of AZI with SH was similar with the thermal behavior of pure AZI and SH. Moreover, intensity of the first peak of AZI was decreased in the physical mixture thermogram which could be explained by dilution effect of polymer, while intensity of the second peak was increased owing to the merging of endothermic peaks between AZI and SH. However, the typical melting endotherms of AZI were disappeared in the thermogram of AZI-SH, suggesting the structure transformation from crystalline form to amorphous form. In addition, the endothermic peak of SH was shifted to 184 °C and several new endothermic peaks appeared among the range from 200 to 225 °C. Based on the results of the thermograms of AZI, SH and AZI-SH, we hypothesized that the carboxyl group of SH could interact with the tertiary amine group of AZI to form the complex of AZI-SH.

### 3.3. XRD Analysis

XRD was performed to confirm the crystalline conversion of SH and AZI. As shown in Figure 2B, only broad spectrum was detected for SH, which was consistent with the amorphous form of this polymer. Several different peaks over a wide range of values were clearly shown in the diffraction of AZI, which demonstrated the crystal structure of AZI. The pattern of physical mixture of AZI and SH was almost similar to that of AZI, which was consistent with the results of DSC. However, the absence of typical peaks of pure AZI in the spectrum of AZI-SH complex suggested the scarcity of crystal structure of AZI. On the other hand, AZI-SH complex showed a novel diffraction peak around 30°, indicating the forming of a new crystalline structure, which might result from an interaction between AZI and SH in the AZI-SH complex.

### 3.4. ^1^H-NMR Analysis

^1^H-NMR analyses were performed to precisely illustrate the interaction between AZI and SH. The spectra of SH, AZI, physical mixture of AZI and SH and AZI-SH complex were shown in Figure 2C. No significant changes of AZI signals were observed in the AZI-SH complex when compared to the spectra of AZI alone, while signals of the physical mixture of AZI and SH were the combination of both compounds. Due to significant signal overlap in most parts of the proton spectrum, the changes could only be obtained in the region between approximately 2.5 and 3.0 ppm. The germinal protons of P1 and P2 (Figure 2D) in AZI had signals at 2.5–3.0 ppm. This peak turned out to be an irregular quartet peak at 2.86 ppm. However, the peak at the same chemical shift was spilt to an irregular triplet peak and an irregular singlet peak in the spectra of AZI-SH. It was speculated that the tertiary amine group of AZI could turn into a positively charged center which would attract electrons and interact with the carboxyl group of SH bearing the negative charge. The results from DSC, XRD and NMR indicated that AZI could interact with SH in mild acid to form a new complex.

### 3.5. Rheological Study

The retention time of different formulations on the eye surface has been reported to be significantly affected by their rheology behaviors [35]. The increased viscosity value can increase the retention time of formulations on the application area especially for topical formulations [36]. However, it has been proved that the addition of water-soluble polymer excipient into eye drops only delayed the elimination of drugs in the first few minutes, but not improve local bioavailability significantly [37]. Therefore, it is necessary to investigate the rheology behaviors of AZI-SH and AZI-HPMC eye formulations to evaluate whether the viscosity of these two formulations affects the retention time of the formulations and the bioavailability of AZI.

The experiments were conducted at the rotational speed ranging from 25 to 220 rpm. As shown in Figure 3, the shearing thinning property indicated the pseudo-plastic rheology of these two formulations. In this flow behavior, an enhanced shearing speed led to an increase of shear stress, which facilitated drugs to be entrapped in the cross-linked network. On the contrary, drugs could be released along the flow when the viscosity decreased due to shearing [38].

On the one hand, the viscosity value decreased with the increase of shearing force when the shearing rate altered in the range of 4.69–49.54 s^−1^ at room temperature (Figure 3). Shear thinning property is always expected since preparations should thin during application and thicken otherwise, for the blinking rate is rapid, ranging from 0.03 s^−1^ during inter-blinking periods to 4250–28,500 s ^−1^ during blinking [39,40]. At the same time, it was further shown that the viscosity of HPMC eye drops was three times higher than that of SH eye drops. A higher viscosity might improve the retention time but may cause discomfort and blurred vision, which leads to poor treatment compliance. However, a lower viscosity might be easily washed by tears, thus resulting in the lower drug efficacy. Viscoelastic fluid, with the advantages that the viscosity becomes high when sheared low and becomes low when sheared quickly, could spread readily on the ocular surface to increase bioavailability and provide a better compliance of patients.

On the other hand, it can be concluded from the comparison of AZI-SH formulation and blank SH that formation of AZI-SH complex will decrease the viscosity of SH, while the viscosity of AZI-HPMC formulation has only decreased slightly with no significant change comparing with the blank HPMC.

### 3.6. Precorneal Pharmacokinetic Study

In vivo precorneal pharmacokinetic studies were carried out to compare the ocular pharmacokinetic profiles of AZI-HPMC and AZI-SH preparations. The concentration-time profiles for AZI after topical instillation of these two kinds of formulations to the rabbit eyes in precorneal are shown in Figure 4. The C_max_ for AZI-SH was 6.59 ± 0.92 mg/mL, 1.2 times higher than that of AZI-HPMC, and the drug concentration of AZI-SH group at every sampling time point was higher than that of AZI-HPMC group. The drug concentration-time data were fitted by compartment model and the results were determined by Akaike’s information criterion [39]. It was finally verified that this variation with time was in accord with two-compartment model, correlation coefficient (*R*) > 0.999, which was consistent with previous studies that the double exponential function was the most similar function to the elimination curve of the drug in the conjunctival sac [41]. In any case, the elimination process always contains the rapid elimination phase and the subsequent slow elimination phase.

The specific pharmacokinetic parameters of AZI in tear fluid after topical administration was shown in Table 2. Parameters based on statistical moment theory (Table 3) for AZI were showing that the area under the curve (AUC) of AZI-SH and AZI-HPMC preparations were 7.30 and 4.60 mg × h/mL, respectively, which indicated a significantly higher drug availability of AZI-SH preparation compared to AZI-HPMC preparation (1.58 times, *p* < 0.01). The mean residence time (MRT) of AZI-SH preparation was 1.56 times higher than that of AZI-HPMC preparation (*p* < 0.05). The results of rheologic study and precorneal pharmacokinetic study suggested a better residence behavior of AZI-SH preparation, despite of a lower viscosity, due to the bioadhesive property of SH which had a high degree of viscoelasticity. The mucus covered by the ocular surface itself is a mixture of mucopolysaccharides, glycoproteins and some other related substances. Therefore, a large number of negatively charged carboxyl groups of SH molecules have a strong interaction with mucus network through hydrogen bonds [42].

The statistic results indicated that the AZI-SH preparation could lead to a slow elimination of AZI from the precorneal area due to the interaction between AZI and SH, which contributed to the increase of the precorneal residence and thus improving the ocular drug availability of AZI.

### 3.7. Safety Evaluation

#### 3.7.1. Cytotoxicity Assay

Two methods, including MTT and NRU assay, were used to evaluate cytotoxicity more comprehensively (Figure 5). These methods are both quantitative assays; the former is related to mitochondrial activity [43] and the latter depends on the uptake of NR into lysosomes [44]. Meanwhile, neutral red has been accepted officially by the French government for the evaluation of cosmetics [45]. The results of the MTT assay indicated that AZI-SH eye drops showed minimal toxicity in the range of 0.05–50 μg/mL, while BZK caused severe cytotoxicity (Figure 5A). Based on the MTT assay, NRU assay was performed with the dosage of AZI-SH eye drops ranging from 0.25 to 80 μg/mL. The results from both assays indicated that the toxicity of eye drops significantly increased with the concentration of BZK increased, suggesting that the toxicity of eye drops was associated with the addition of BZK, which has been recognized as the major toxic factor in eye drops [46]. BZK also showed a concentration- and time-dependent cytotoxicity, indicating that a higher concentration or a longer exposure of BZK caused much severe cytotoxicity [47]. The use of macromolecules such as SH and polycarbophil significantly decreased the toxicity of BZK, evidenced in Table 4 by the significantly higher half-maximal inhibitory concentration (IC_50_)(5.71 times) and ED_50_ (4.19 times) of AZI-SH eye drops compared to that of BZK solution. The IC_50_ and ED_50_ of AZI-SH eye drops was 1.17 times and 1.24 times higher than that of commercial AZI eye drops, suggesting a better protective effect of SH against BZK than polycarbophil. The findings were consistent with the results obtained from the MTT assay against HCE-2 cells (Appendix A). It has been reported that BZK could overproduce hydroperoxide and superoxide anion, leading to an oxidative stress [24]. However, the cationic quaternary ammonium of BZK which is harmful to eye tissues can be neutralized by the negative charges of SH. Meanwhile, SH contains a large number of hydroxyl groups, which can absorb reactive oxygen species produced by BZK. Therefore, SH is an effective agent against BZK-induced cytotoxicity, which is accordance with previous findings [24,48].

#### 3.7.2. In Vivo Eye Irritation Assessment in Rabbits

Evaluation and classification standard of rabbit eye irritation was referred to the new drug preclinical research guidelines issued by the Ministry of Health in China. During the whole processes, no deaths occurred by instillation and no obvious changes in rabbits’ behavior were observed. The scores of irritation reactions of cornea, conjunctiva and iris were assessed. The comprehensive average score of the eye irritation response of the left eye given sterile saline was 0.25, the average score of the eye irritation response of the right eye given AZI-SH eyedrops was 0.25, indicating that the product had no eye irritation. The histological images of the major tissues (cornea, conjunctiva, sclera and iris) using hematoxylin and eosin staining are shown in Figure 6. Similar to the saline group, no noticeable tissue damage and no significant difference of the irritation scores between the treatment and control groups were observed. The results indicated that AZI eye drops did not show any irritation to rabbits’ eyes.

#### 3.7.3. Acute Toxicity Test in Rabbits

For the acute toxicity test, the pupil changes of rabbits treated with a high concentration of AZI-SH eye drops were monitored. The phenomenon of pupil variation was listed in Appendix A. No significant changes of pupil were observed in both treatment and control group since. No obvious congestion was detected in the conjunctiva (palpebral conjunctiva, fornical conjunctiva and bulbar conjunctival) and iris. No cloudy symptom and ulcer of cornea and no abnormal secretions were observed. During the observation period, the AZI did not show any damages to rabbit local eye tissues or other tissues, which indicated that AZI eye drops would not cause any local or systemic toxicity.

## 4. Conclusions

A new AZI-SH eye drops formulation was developed, in which AZI and SH could form a complex under the mild acidic condition. By the biological adhesion advantage of SH, this novel ocular preparation could extend precorneal residence and improve the ocular drug availability of AZI. In addition, the developed AZI-SH eye drops showed a good safety profile both in vitro and in vivo. Therefore, our findings indicated that this SH-based AZI eye drops was a promising formulation for topical drug delivery and ophthalmic therapy.

## Figures and Tables

**Figure 1 pharmaceutics-11-00183-f001:**
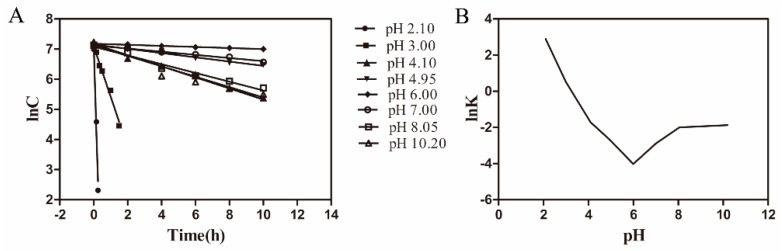
Degradation kinetics of azithromycin. **A**: Apparent first-order plot of azithromycin (AZI) hydrolysis at various pH values. Temperature was 75 °C and ionic strength (*I*) = 0.3. **B**: the hydrolysis of AZI at various pH, 75 °C and *I* = 0.3.

**Figure 2 pharmaceutics-11-00183-f002:**
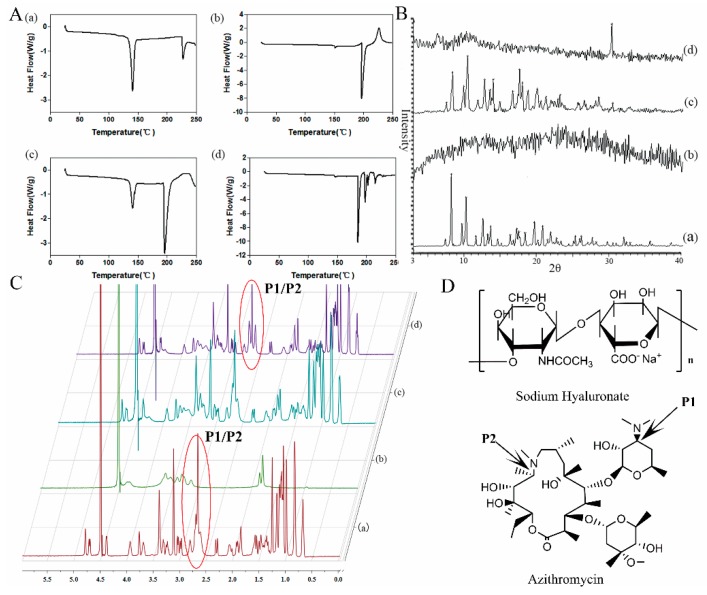
Illustration of interaction between AZI and sodium hyaluronate (SH). **A**: Differential Scanning Calorimetry (DSC) thermograms; **B**: X-ray diffraction (XRD) patterns; **C**: ^1^H-Nuclear Magnetic Resonance (^1^H-NMR) spectra. (a) AZI, (b) SH, (c) physical mixture of AZI+SH and (d) AZI-SH complex. **D**: structure of sodium hyaluronate and azithromycin.

**Figure 3 pharmaceutics-11-00183-f003:**
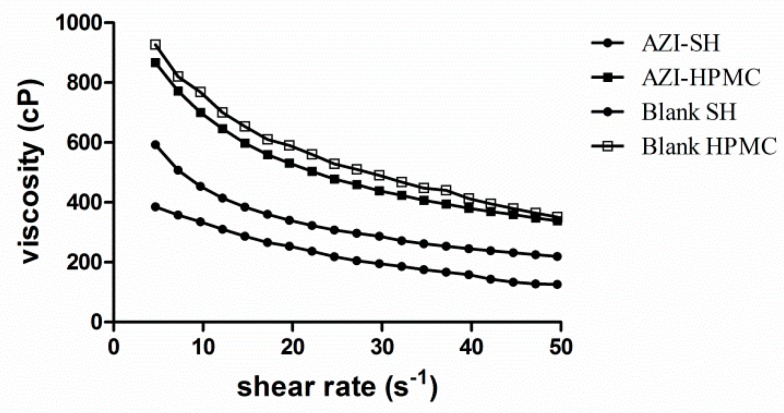
Comparison of the viscosity between AZI-SH formulation, AZI-HPMC formulation, blank SH and blank HPMC.

**Figure 4 pharmaceutics-11-00183-f004:**
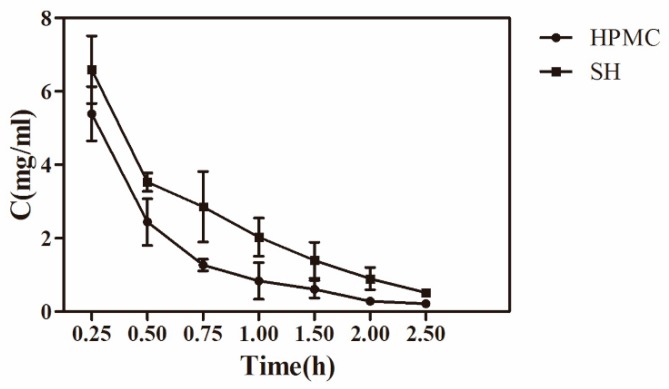
Concentration-time profiles of AZI in tear fluid after instillation of HPMC preparation and SH preparation in conscious rabbits (*n* = 5).

**Figure 5 pharmaceutics-11-00183-f005:**
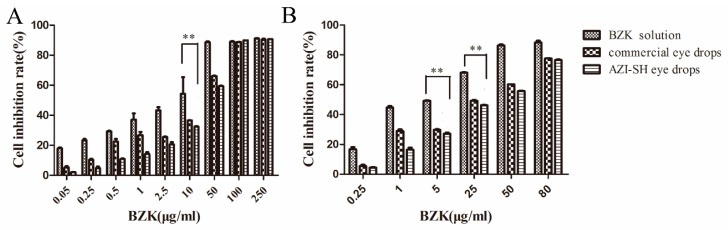
Cell viability against 3T3-L1 cells determined by MTT assay (**A**) and Neutral red uptake (NRU) assay (**B**). (** *P* < 0.01).

**Figure 6 pharmaceutics-11-00183-f006:**
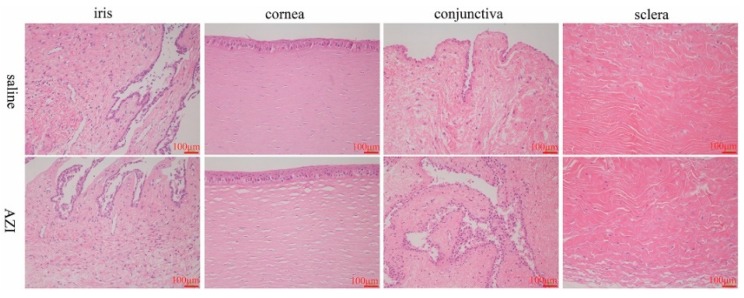
Evaluation of eye irritation by histopathological examinations of iris, cornea, conjunctiva and sclera.

**Table 1 pharmaceutics-11-00183-t001:** Rate constants (*K*) for the degradation of AZI in aqueous solution at different pH values (1–10).

pH	*K* (h^−1^)	ln*K*	*R*
2.1	18.1000	2.90	0.9849
3.0	1.6747	0.52	0.9955
4.1	0.1805	−1.71	0.9965
4.95	0.0689	−2.68	0.9947
6.0	0.0179	−4.02	0.9953
7.0	0.0557	−2.89	0.9918
8.05	0.1354	−2.00	0.9849
10.2	0.1534	−1.87	0.9852

Temperature was 75 °C, ionic strength (*I*) = 0.3, and *R* is the linear correlation coefficient.

**Table 2 pharmaceutics-11-00183-t002:** Pharmacokinetics parameters of AZI in tear fluid after topical administration in the conscious rabbits (*n* = 5).

Pharmacokinetics Parameters	HPMC Group	SH Group
A (μg/mL)	12.39	31.35
α (1/h)	4.42	10.83
B (μg/mL)	1.58	5.34
β (1/hr)	0.81	0.90
V(c) (mg)/(mg/mL)	71.60	13.63
T_1/2α_ (h)	0.16	0.06
T_1/2β_ (h)	0.86	0.77
K_21_ (1/h)	1.22	2.3438
K_10_ (1/h)	2.94	4.15
K_12_ (1/h)	1.08	5.24
AUC (mg/mL) × h	4.76	8.85
CL(s) μg/h/(mg/mL)	210.18	56.52

**Table 3 pharmaceutics-11-00183-t003:** Parameters based on statistical moment theory for AZI in tear fluid after topical administration in the conscious rabbits (*n* = 4).

Pharmacokinetics Parameters	HPMC Group	SH Group
Area under the curve (AUC)(mg × h/mL)	4.60 ± 0.51	7.30 ± 0.42
Area under the moment curve (AUMC)(mg × h^2^/mL)	3.16 ± 0.52	7.77 ± 0.58
Mean residence time (MRT)(h) ***	0.69 ± 0.03	1.07 ± 0.05
Variance of the residence time (VRT)(h × h)	1.07 ± 0.02	1.61 ± 0.02

(*** *p* < 0.05).

**Table 4 pharmaceutics-11-00183-t004:** IC50 and ED50 values of BZK solution, AZI-SH eye drops and commercial AZI eye drops in MTT assay and NRU assay.

Groups	IC50 (μg/mL)	ED50 (μg/mL)
BZK solution	5.09 ± 0.96	9.69 ± 5.66
AZI-SH eye drops	29.07 ± 1.87	40.57 ± 1.36 *
commercial AZI eye drops	24.80 ± 2.06	32.79 ± 0.48 *

(* *P* < 0.01).

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
