# Peer review of "Preparation and Evaluation of Topically Applied Azithromycin Based on Sodium Hyaluronate in Treatment of Conjunctivitis"

_pharmaceutics, 2019, doi:10.3390/pharmaceutics11040183_

Round 1

Reviewer 1 Report

The manuscript presents preparation and characterization of  Azithromycin (AZI) formulation based on sodium hyaluronate (SH) intended for topical administration to treat conjunctivitis. Authors included cytotoxicity evaluation and pharmacokinetics.

Comments:

-        In Abstract, authors stated “AUC of AZI in SH-containing formulation was 1.58-fold higher (p<0.01) than that in HPMC-containing formulation due to the interaction between the amine group of AZI and the carboxyl group of SH”. Not clear how the interaction would improve bioavailability.

-        Recommend to rephrase the sentences such as “it was waste of drugs by oral administration.”; “worse patient compliance” etc.

-        In introduction authors stated “ the binding sites for hyaluronate have been identified on corneal endothelium, which contributes to the bioadhesive effects of SH with eye tissue”.  Did authors refer to pre-corneal adhesion? If the binding sites are on endothelium, how they will contribute to precorneal adhesion?.

-        Did authors determine viscosity of both formulations?

-        There is published literature on pH dependent degradation kinetics of AZI. Significance and relevance of including this study is not clear.

-        Section 2.2.10 states “Values of P<0.05 were regarded as statistical significant”. Figure 5, figure legend states (*p>0.5). Per visual observation of the graph, it appears that there is no significant difference between commercial eye drops vs AZI-SH eye drops at 10 ug/ml (MTT assay) and at 5 & 25 ug/ml (NRU assay).  Verify the statistical significance.

-        Recommend to replace Figure 2 with high resolution one.

-        Recommend to include the references for calculating pharmacokinetic parameters for drug concentrations in tears. Not clear how significant reported AUCs are with respect to drug concentrations at target site.

-        Figure 5 and Figure S1: the X-axis label is BZK (ug/ml). Was concentration of BZK changed in the commercial and AZI-SH eye drops? Please clarify.

-        Manuscript needs to be edited to correct typos and language errors.  

Author Response

Figure 3. Comparison of the viscosity between AZI-SH formulation, AZI-HPMC formulation, blank SH and blank HPMC.

-        There is published literature on pH dependent degradation kinetics of AZI. Significance and relevance of including this study is not clear.

Response: Although the study of degradation kinetics of AZI has already been widely reported, it is also necessary to illustrate the stability of AZI over a range of pHs throughout our investigation. In this paper, degradation kinetics of AZI is important for the selection of pH of AZI-SH eyedrops. Indeed, AZI was most stable at pH6.0, while pH 5.5 was chosen based on the optimization of AZI-SH eyedrops using Box-Behnken Design (data were published in other place). During these designs, three main significant factors, including the pH of the eye dropsthe molecular weight of SH and the mass concentration of SH were taken into considerationand Box-Behnken design was used with pharmacokinetics parameters as the indexes to determine the optimum formulationThe optimum formulation of azithromycin eye drops: pH 5.5molecular weight of SH was 1.5 million, and the mass concentration of SH was 0.6 g/ (100 ml) [1].

-        Section 2.2.10 states “Values of P<0.05 were regarded as statistical significant”. Figure 5, figure legend states (*p>0.5). Per visual observation of the graph, it appears that there is no significant difference between commercial eye drops vs AZI-SH eye drops at 10 ug/ml (MTT assay) and at 5 & 25 ug/ml (NRU assay).  Verify the statistical significance.

Response: Statistical significance of commercial eye drops vs AZI-SH eye drops at 10 ug/ml (MTT assay) and at 5 & 25 ug/ml (NRU assay) was rechecked and no significant difference existed. “*P>0.5” was meant to show that there was no significant difference between these groups, which was redundant. So, we have removed these markers in Figure 5.

Figure 5. Cell viability against 3T3-L1 cells determined by MTT assay (A) and NRU assay (B). (**P<0.01)

-        Recommend to replace Figure 2 with high resolution one.

Response: The resolution of Figure 2 has been improved to 600dpi.

Figure 2. Illustration of interaction between AZI and SH. A: DSC themrograms; B: XRD patterns; C: 1H-NMR spectra. (a) AZI, (b) SH, (c) physical mixture of AZI+SH and (d) AZI-SH complex. D: structure of sodium hyaluronate and azithromycin.

-        Recommend to include the references for calculating pharmacokinetic parameters for drug concentrations in tears. Not clear how significant reported AUCs are with respect to drug concentrations at target site.

Response: Rapid loss of the instilled solution due to lacrimal drainage through the drainage apparatus has considerable effect on the bioavailability of ophthalmic drugs [2]. This results in short contact time between drug and cornea, leading to reduced drug availability. In the previous studies, pharmacokinetics parameters in tear fluid, cornea, conjunctiva, aqueous humor, and plasma were widely investigated [3, 4]. In the current study, we mainly focused on the precorneal retention ability of AZI-SH eye drops. So, our target site was precorneal and drug concentration in the tears was taken into consideration. Our study is still going on, drug distribution in other parts of eye tissues will be published in our later work.

-        Figure 5 and Figure S1: the X-axis label is BZK (ug/ml). Was concentration of BZK changed in the commercial and AZI-SH eye drops? Please clarify.

Response: The concentration of BZK in the commercial eye drops was 0.003%, which was 0.001% of that in the AZI-SH eye drops. BZK was the main reason led to eye irritation or ocular adverse reaction. So, BZK concentration was used as the factor to evaluate cytotoxicity.

-        Manuscript needs to be edited to correct typos and language errors. 

Response: we have rechecked the manuscript.

References:

[1] Y. Sun, Q. Chen, C. Yin, J. Tu, Y. Shen, X. Wan, Formulation optimization and preparation process investigation of azithromycin eye drops, Chinese Journal of Hospital Pharmacy 38(7) (2018) 701-707.

[2] S.S. Chrai, T.F. Patton, A. Mehta, J.R. Robinson, Lacrimal and Instilled Fluid Dynamics in Rabbit Eyes, Journal of Pharmaceutical Sciences 62(7) (2010) 1112-1121.

[3] E.K. Akpek, J. Vittitow, R.S. Verhoeven, K. Brubaker, T. Amar, K.D. Powell, J.L. Boyer, C. Crean, Ocular surface distribution and pharmacokinetics of a novel ophthalmic 1% azithromycin formulation, Journal of Ocular Pharmacology & Therapeutics the Official Journal of the Association for Ocular Pharmacology & Therapeutics 25(5) (2009) 433.

[4] B. Liu, L. Ding, X. Xu, H. Lin, C. Sun, L. You, Ocular and systemic pharmacokinetics of lidocaine hydrochloride ophthalmic gel in rabbits after topical ocular administration, Eur J Drug Metab Pharmacokinet 40(4) (2015) 409-15.

Reviewer 2 Report

As a general comment, the English should be improved in order to make the manuscript easier to follow.

Introduction:

1.       The final sentence belongs in a “Conclusion” and not the “Introduction”.

Materials

2.       Section 2.2.8, the choice of pharmacokinetic model should be given here.

Results and Discussion

3.       I do not see the relevance of the DSC.  The authors have effectively made two gels and their viscosity may have an effect on the residence time of the drug at the eye surface, as reflected by the concentrations found in the tear fluid.

4.       What do the pharmacokinetic parameters really mean ?  The concentrations presented are those in the tear fluid, not those found in any of the ocular compartments. Hence, I do not think that the use of the term bioavailability is correct either since the drug has not been quantified in the eye.  The data simply present the residence/elimination of the drug from the eye surface and the concentrations in the tear fluid.

Author Response

Dear editors and reviewers,

On behalf of all my co-authors, I would like to thank you for providing us with the opportunity to revise our manuscript entitled "Preparation and evaluation of topically applied azithromycin based on sodium hyaluronate in treatment of conjunctivitis" (ID: pharmaceutics-456672). We really appreciate and benefited from the constructive comments provided by both editors and reviewers. We have carefully reviewed the comments and revised the manuscript accordingly.

We hope that our responses can properly address all of the concerns and the revised version is now suitable for publication in Pharmaceutics.

Listed below are our point-by-point responses to reviewer 2 comments.

Introduction:

1.       The final sentence belongs in a “Conclusion” and not the “Introduction”.

Response: The last sentence has been removed from introduction.

Materials

2.       Section 2.2.8, the choice of pharmacokinetic model should be given here.

Response: In section 3.6, compartment model and non-compartment model (statistical moment theory) were used to analyze pharmacokinetic parameters. As for the compartment model, the results were determined by Akaike’s information criterion. It was finally verified that this variation with time was in accord with two-compartment model, correlation coefficient(R) > 0.999. So, two-compartment model was chose to calculate the pharmacokinetic parameters.

Results and Discussion

3.       I do not see the relevance of the DSC.  The authors have effectively made two gels and their viscosity may have an effect on the residence time of the drug at the eye surface, as reflected by the concentrations found in the tear fluid.

Response: DSC was conducted to investigate the interaction between AZI and SH, which was a commonly used method to study the transformation of crystal structure. AZI, SH, physical mixture of AZI and SH, and the complex of AZI-SH were examined in the present study. From the results of DSC, the typical melting endotherms of AZI disappeared in the thermogram of AZI-SH, suggesting the structure transformation from crystalline form to amorphous form. In addition, the endothermic peak of SH was shifted to 184 °C and several 261 new endothermic peaks appeared among the range from 200 °C to 225 °C. These patterns can be used to illustrate the formation of AZI-SH complex owing to the interaction of AZI and SH, which will endow AZI-SH eye drops with the bioadhesive property of SH to improve the bioavailability.

4.       What do the pharmacokinetic parameters really mean?  The concentrations presented are those in the tear fluid, not those found in any of the ocular compartments. Hence, I do not think that the use of the term bioavailability is correct either since the drug has not been quantified in the eye.  The data simply present the residence/elimination of the drug from the eye surface and the concentrations in the tear fluid.

Response: Rapid loss of the instilled solution due to lacrimal drainage through the drainage apparatus has considerable effect on the bioavailability of ophthalmic drugs [1]. It is expected that 99 % of the drug is lost from the precorneal area. This results in short contact time between drug and cornea, leading to reduced drug availability. In the previous studies, pharmacokinetic parameters in tear fluid, cornea, conjunctiva, aqueous humor, and plasma were widely investigated [2, 3]. In our current study, we mainly focused on the precorneal residence ability of both eye drops. So, pharmacokinetic parameters in the tears were taken into consideration.

References:

[1] S.S. Chrai, T.F. Patton, A. Mehta, J.R. Robinson, Lacrimal and Instilled Fluid Dynamics in Rabbit Eyes, Journal of Pharmaceutical Sciences 62(7) (2010) 1112-1121.

[2] E.K. Akpek, J. Vittitow, R.S. Verhoeven, K. Brubaker, T. Amar, K.D. Powell, J.L. Boyer, C. Crean, Ocular surface distribution and pharmacokinetics of a novel ophthalmic 1% azithromycin formulation, Journal of Ocular Pharmacology & Therapeutics the Official Journal of the Association for Ocular Pharmacology & Therapeutics 25(5) (2009) 433.

[3] B. Liu, L. Ding, X. Xu, H. Lin, C. Sun, L. You, Ocular and systemic pharmacokinetics of lidocaine hydrochloride ophthalmic gel in rabbits after topical ocular administration, Eur J Drug Metab Pharmacokinet 40(4) (2015) 409-15.

Reviewer 3 Report

This paper is a very interesting study.

I would like to underline some aspects that could be improved:

1. line 91 - the detailed experimental procedures of degradation kinetics of AZi are not found in the Supplementary information. Please add this information.

2. A more appropiate term is "rotational speed" instead "angular velocities". See lines 133, 302 ...

3. The resolution of Figure 1 A and B needs improvement.

4. Figure 2 A-C are not very clear and difficult to read. Please improve this figure.

5. As a suggestion, for a better visualization of the results, in Figure 3 you could replace the blank representation with empty symbols.

6. For future studies I suggest to quantify the rheological behaviour using Power law model that express the relation between viscosity and shear rate.

Author Response

Dear editors and reviewers,

On behalf of all my co-authors, I would like to thank you for providing us with the opportunity to revise our manuscript entitled "Preparation and evaluation of topically applied azithromycin based on sodium hyaluronate in treatment of conjunctivitis" (ID: pharmaceutics-456672). We really appreciate and benefited from the constructive comments provided by both editors and reviewers. We have carefully reviewed the comments and revised the manuscript accordingly.

We hope that our responses can properly address all of the concerns and the revised version is now suitable for publication in Pharmaceutics.

Listed below are our point-by-point responses to reviewer 3 comments.

1. line 91 - the detailed experimental procedures of degradation kinetics of AZi are not found in the Supplementary information. Please add this information.

Response: The detailed experimental procedures of degradation kinetics of AZI have already been presented in supplementary information as follows:

A stock solution containing 1% AZI was used to investigate the degradation kinetics of AZI. The stock solution (1 ml) was separately diluted to 10 ml using phosphate buffer solutions with different pH values (1, 3, 5, 7, 10) and incubated at 75 °C for 2 h, 4 h, 6 h, 12 h and 24 h. The incubate solutions (0.5 ml) was then diluted ten times with the mobile phase to obtain test solutions. The actual pH values of each test solution were monitored. Moreover, the concentration of AZI (CAZI) at each time points was quantified by HPLC-UV. Then, lnCAZI was plotted against time (t) to analyze the apparent hydrolysis rate constant (K) and the kinetic order of degradation reactions under different pH. Subsequently, the pH where AZI was the most stable (pHm) was obtained from the lnk-pH curve, which was located at the lowest point of this curve.

2. A more appropiate term is "rotational speed" instead "angular velocities". See lines 133, 302 ...

Response: All of the “angular velocities” have been replaced by “rotational speed” (lines 131, 133 and 302).

3. The resolution of Figure 1 A and B needs improvement.

Response: The resolution of Figure 1 has been improved to 600dpi.

Figure 1. Degradation kinetics of azithromycin. A: Apparent first-order plot of AZI hydrolysis at various pH values. Temperature was75°C and ionic strength (I) = 0.3. B: The hydrolysis of AZI at various pH, 75 °C and I = 0.3.

4. Figure 2 A-C are not very clear and difficult to read. Please improve this figure.

Response: The resolution of Figure 2 has been improved to 600dpi.

Figure 2. Illustration of interaction between AZI and SH. A: DSC themrograms; B: XRD patterns; C: 1H-NMR spectra. (a) AZI, (b) SH, (c) physical mixture of AZI+SH and (d) AZI-SH complex. D: structure of sodium hyaluronate and azithromycin.

5. As a suggestion, for a better visualization of the results, in Figure 3 you could replace the blank representation with empty symbols.

Response: We have changed the markers of two blank groups for a better visualization.

Figure 3. Comparison of the viscosity between AZI-SH formulation, AZI-HPMC formulation, blank SH and blank HPMC.

6. For future studies I suggest to quantify the rheological behavior using Power law model that express the relation between viscosity and shear rate.

Response: Thanks for your generous advice. Our research is still continuing and we will furtherly investigate on this issue and the Power law model will be displayed in our later papers.

Round 2

Reviewer 1 Report

Authors addressed comments raised by reviewers. No further comments.

Author Response

Thank you very much for evaluating our manuscript.  We really appreciate the comments that you  have made on the manuscript, which are greatly helpful to improve the quality of our work. 

Reviewer 2 Report

Given that the drug concentrations are presented in the tear fluid - what is the physiological justification for using a 2-compartment model for the PK analyis ? Please include the rationale in the manuscript.

Although it is clear that a short ocular residence time will limit bioavailability in the eye, I still do not consider that bioavailability is the correct term to describe the concentrations in the tear fluid - the drug is not bioavailiable

Author Response

Dear reviewer,

Thank you very much for evaluating our manuscript.  We really appreciate the comments that you have made on the manuscript, which are greatly helpful to improve the quality of our work. Listed below are our point-by-point responses

Given that the drug concentrations are presented in the tear fluid - what is the physiological justification for using a 2-compartment model for the PK analyis ? Please include the rationale in the manuscript.

Response: Several studies have investigated precorneal clearance of mucoadhesive materials from eyes and radiolabeling method was used. The remaining radioactivity of cornea, the inner canthus and the lacrimal duct was detected by a gamma camera. The clearance kinetics was fitted to a biphasic equation with an initial rapid phase followed by a much slower basal phase [1, 2].

In this study, we mainly focused on the treatment of conjunctivitis, which required the plenty amount of drug in the anterior segment of the eyes. So drug concentration in tear film could provide a reflection of amount of drug that could be accessible to the conjunctiva.

Of course, it is better to use the amount of drug in the conjunctiva, which we will study later.

Although it is clear that a short ocular residence time will limit bioavailability in the eye, I still do not consider that bioavailability is the correct term to describe the concentrations in the tear fluid - the drug is not bioavailiable

Response: We have read several papers about the pharmacokinetics investigation of ophthalmic formulations. The bioavailability was studied by analyzing drug concentration not only in tears but also in many eye tissues. So, it is a good suggestion that bioavailability might not be quite suitable. So, bioavailability was replaced by drug availability.

Reference:

[1] A.M. Durrani, S.J. Farr, I.W. Kellaway, Influence of molecular weight and formulation pH on the precorneal clearance rate of hyaluronic acid in the rabbit eye, International Journal of Pharmaceutics 118(2) (1995) 243-250.

[2] A.M. Durrani, S.J. Farr, I.W. Kellaway, Precorneal clearance of mucoadhesive microspheres from the rabbit eye, Journal of Pharmacy & Pharmacology 47(7) (2011) 581-584.

Round 3

Reviewer 2 Report

Thank you for the modification of the manuscript and the clarification of the terms.